Design and simulation-based evaluation of a novel tracheostomy high-flow therapy device interface: a preclinical study

Hou Anna
Mi Song
Jiao Fengwei
Zhang Liming cyyyzlm@sina.com
Tong Zhaohui tongzhaohuicy@sina.com
Department of Respiratory and Critical Care Medicine, Beijing Institute of Respiratory Medicine and Beijing Chao-Yang Hospital, Capital Medical University , Beijing , China
Anson Lesley
Electronic publication date: 2025 Dec 16
Publication date: 2025
Volume: 13
Electronic Location ID: e20445
Received 2025 Mar 26; Accepted 2025 Oct 31
Copyright: ©2025 Hou et al.
Copyright year: 2025
Copyright holder: Hou et al.
License: This is an open access article distributed under the terms of the Creative Commons Attribution License, which permits unrestricted use, distribution, reproduction and adaptation in any medium and for any purpose provided that it is properly attributed. For attribution, the original author(s), title, publication source (PeerJ) and either DOI or URL of the article must be cited.
License URL: https://creativecommons.org/licenses/by/4.0/

Keywords: High-flow tracheal oxygen, Novel high-flow tracheal interface, Positive end-expiratory pressure, Functional residual capacity, Tidal volume

Funding: The authors received no funding for this work.

==============================
Background

High-flow tracheal oxygen (HFTO) does not generate significant positive end-expiratory pressure (PEEP), which may limit its clinical applicability. We designed a novel high-flow tracheal interface (nHFTI) and compared the PEEP and other respiratory parameters it generated with those of the traditional high-flow tracheal interface (tHFTI) across a range of flow rates.

Methods

We conducted a randomized crossover bench study. A size 8 cuffed tracheostomy tube was connected to the high-flow therapy (HFT) device using two different interfaces. A high-fidelity lung simulator was used to model normal lung mechanics. Gas flow rates were set from 10 to 80 L/min at a gradient of 10 L/min, and the sequence was randomized. At each flow setting, the following parameters were recorded: PEEP, peak inspiratory pressure (PIP), peak expiratory pressure (PEP), end-inspiratory transpulmonary pressure (Ptp-EI), end-expiratory transpulmonary pressure (Ptp-EE), end-inspiratory cardiac pressure (PEIC) and end-expiratory cardiac pressure (PEEC), functional residual capacity (FRC), tidal volume (Vt) and fraction of inspired oxygen (FiO2).

Results

As the HFT flow rate increased, PEEP increased significantly with both the tHFTI and nHFTI (P < 0.001). Compared with the tHFTI, the nHFTI generated significantly higher PEEP across most flow rates (e.g., 3.4 vs. 1.0 cmH2O at 40 L/min; 7.3 vs. 3.1 cmH2O at 60 L/min, nHFTI vs. tHFTI, mean values). The nHFTI also generated higher PIP, PEP, Ptp-EI, Ptp-EE, PEIC, and PEEC, and resulted in a greater increase in FRC (P < 0.010). Compared with the tHFTI, the nHFTI maintained a relatively constant FiO2 and slightly reduced Vt at the same flow rate level.

Conclusions

The nHFTI generated a modest level of PEEP, increased FRC and slightly reduced Vt at flow rates of 40–60 L/min. This physiological profile suggests potential for maintaining lung expansion and improving cardiac function in heart failure. Further in vivo studies are needed to evaluate the clinical impact of nHFTI-mediated PEEP augmentation and determine whether it can lead to improved clinical outcomes, particularly in patients requiring prolonged mechanical ventilation during the weaning process.

Introduction

Tracheostomy is performed in 10–15% of critically ill patients requiring prolonged mechanical ventilation to facilitate the weaning process (Abe et al., 2018; Aquino Esperanza, Pelosi & Blanch, 2019), with increasing adoption over the past decade (Esteban et al., 2013). Unlike conventional oxygen therapy, high-flow tracheal oxygen (HFTO) delivers inhaled gas with stable temperature, humidity, and oxygen concentration. This maintains mucociliary clearance, ensures airway hydration, and promotes normothermia. Case reports document successful weaning with HFTO (Mitaka et al., 2018; Lionello et al., 2023; Lytra et al., 2024).

However, as an upper airway-bypassing system, HFTO generates minimal positive end-expiratory pressure (PEEP). Limited studies (Moorhouse et al., 2015; Mitaka et al., 2018; Chen et al., 2019; Natalini et al., 2019) have quantified the PEEP delivered by HFTO, which is no more than 2 cmH2O—insufficient to improve respiratory and cardiac function. This evidence-practice gap may limit the clinical adoption of HFTO in patients with tracheostomy.

Chen et al. (2019) modified the HFTO system by adding a 5-cmH2O/L/s resistor to the expiratory port and found it induced an elevation in airway pressure. We hypothesized that adding an exhalation valve to the exhalation port of a traditional interface would significantly increase PEEP levels beyond baseline, and accordingly designed a novel HFTO interface (Chinese Utility Model Patent ZL2024 2 0280842.6). We conducted a randomized crossover study to evaluate the effects of the new interface and the traditional interface on PEEP and other respiratory physiological indicators across different flow rates.

Materials and Methods

We conducted an in vitro laboratory experiment to quantify and characterize the PEEP generated by high-flow therapy (HFT). This methodology was implemented to specifically assess HFT’s isolated mechanical contribution to PEEP without physiological interference. The experiment was undertaken in February 2024. Ethics approval was not required for this study, as it involved no human participants, animals, or biological materials, and was classified as non-biomedical device research under institutional guidelines.

The apparatus involved in the benchtop experiment (Fig. 1)

HFT device:

Figure 1 The apparatus involved in the benchtop experiment.

(A) Traditional high-flow tracheal interface (tHFTI); (B) novel high-flow tracheal interface (nHFTI); B1, exhalation port; B2, inhalation port; B3, patient port; B4, spare port; B5 and B6, exhalation valve; (C) comparison of nHFTI and tHFTI; (D) model with the tHFTI-tracheostomy tube-simulated trachea; (E) model with the nHFTI-tracheostomy tube-simulated trachea; (F) HFLS-simulated trachea-tracheostomy tube-HFTI-HFT device model; (a) High-fidelity lung simulator (HFLS); (b) simulated trachea; (c) tracheostomy tube; (d) high-flow tracheal interface (HFTI); (e) HFT Device; (f) humidifier; (g) non-invasive ventilator (NIV); (h) simplified test lung; (i) platform outlet valve.

The Micomme HFT Device (OH-70C, Hunan, China) and its matching heated delivery tube were selected. Gas flow rates ranged from 10 to 80 L/min, with oxygen concentration adjustable from 21% to 100% and temperature from 29 °C to 37 °C. Its advantage lies in the built-in accurate oxygen concentration monitoring module, which can accurately monitor the oxygen concentration without loading oxygen batteries and other consumables.

High-flow tracheal interface (HFTI):

The traditional HFTI (tHFTI, HFC-09-P-T, Hunan, China, Fig. 1A) adopts the tracheal interface configured by Micomme HFT Device (OH-70C, Hunan, China).

A novel HFTI (nHFTI) has been developed and patented. The core innovation of nHFTI is a 7.5-mm diameter exhalation valve integrated at the exhalation port to increase exhalation resistance. The nHFTI features an 80-mm-long four-way tubular structure with dimensions comparable to the tHFTI. Its ports comprise:

• Exhalation port (ID: 13 mm) with integrated valve

• Inhalation port (ID: 13 mm)

• Patient port (ID: 15 mm)

• Sealed spare port (ID: 13 mm; used as an exhalation end during bronchoscopy or suctioning procedures)

The exhalation port of the HFTI has an internal diameter of 7.5 mm, comparable to that of a standard adult endotracheal tube. The pathway is designed to be short and straight to minimize secretion retention. The interface is intended for use with a commercial high-flow device equipped with real-time pressure monitoring and high-pressure alarms, which detect and alert clinicians to any obstruction in the circuit.

The 3D-printed nHFTI is shown in Fig. 1B and comparison of the two is shown in Fig. 1C.

Tracheostomy tube and simulated trachea:

A size 8 cuffed Shiley tracheostomy tube (313-80, Covidien LLC, Downey, CA, USA) was used. A 100-mm-long ventilator tube with a 22-mm inner diameter was employed to simulate the trachea of an average adult.

High-fidelity lung simulator:

A high-fidelity lung simulator (HFLS, TestChest®, ORGANIS GmbH, Switzerland, Software version 5.07) was used as a reference tool to simulate normal human lungs in the benchtop study. The TestChest® is a full physiologic artificial lung that can simulate gas exchange and hemodynamic responses of the healthy and pathological adult lungs. The device was calibrated according to the manufacturer’s description.

Non-invasive ventilator (NIV):

The Yi’an VG55 NIV (Hebei Yi’an Aomei Medical Equipment Co., Ltd, China) was used to provide a constant airflow, simulating HFT output through a heating and humidification device to maintain normal HFT operation and prevent humidified gases from entering and damaging the simulated lung.

Establish an HFLS-based trachea-tracheostomy tube-HFTI-HFT device model (Figs. 1D–1F)

The simulated trachea was connected to the airway interface of the HFLS. The tracheostomy tube was inserted into the simulated trachea, and its cuff was inflated with 10 mL of air. The seal was checked to confirm the absence of air leakage. The air supply tube of the HFT device was directly connected to the tracheostomy tube via the HFTI, bypassing the heating and humidification device. This configuration delivered dry gas output from the HFT device directly to the HFLS through the HFTI, preventing humidified gas from damaging the simulated lung and affecting test results.

However, bypassing the heated humidifier could trigger alarms or cause malfunction in the HFT device. To maintain normal operation, we connected the air supply tube of the NIV to the inlet of the heated humidifier, and the outlet was connected to a simplified test lung—equipped with a platform outlet valve on the adjacent tubing—via a bypass circuit. The NIV was then activated in CPAP mode at 6 cmH2O to generate a constant airflow through the heated humidifier. This simulated the typical output of the HFT system and ensured stable operation of the device during testing. The gas flow pathways are shown in Fig. 2.

Figure 2 Schematic representation of gas flow pathways in unmodified HFT (upper panel) and in the modified setup of this study (lower panel).

HFT, high-flow therapy; HFTI, high-flow tracheal interface; NIV, non-invasive ventilator.

Parameter settings

HFLS parameter settings:

Under standard temperature, pressure and dry (STPD) conditions, the simulated lung was set to normal breathing mode, with a total respiratory compliance of 100 mL/hPa, to simulate normal human spontaneous breathing. The respiratory rate (RR) was set to 10 breaths/min to minimize the risk of air trapping.

HFT device settings:

For HFT device settings, flow rates were set from 10 L/min up to its maximum set limit of 80 L/min at a flow gradient of 10 L/min with a fraction of inspired oxygen (FiO2) of 0.50 and a temperature of 29 °C.

Study procedures and parameters

A randomized crossover design was used during HFLS operation with high-flow oxygen delivered via the tracheostomy tube using either tHFTI or nHFTI at varying flow rates. Flow rates were adjusted from 10 to 80 L/min in 10 L/min increments, and the sequence was randomized.

Under each condition, HFT was delivered via the 8.0 ID tracheostomy tube using either tHFTI or nHFTI, assigned randomly and applied sequentially.

All measurements were taken at steady state after a 10-minute equilibration period at each flow level. Pressure, flow and volume signals over the last 1 min of each phase were displayed continuously and saved in a laptop at a sample rate of 200 Hz for further analysis. A 5-min washout period was implemented between measurements to ensure no carry-over effect between interventions.

Data were recorded in spreadsheets and displayed graphically using Microsoft Excel. Most parameters were extracted directly from the parameter-time graphs. For each flow setting, a record was taken once every minute for a total of 10 records, and the mean and standard deviation (SD) were calculated.

The following parameters were collected: tidal volume (Vt); functional residual capacity (FRC); PEEP—defined as the average pressure during the last 0.1s of expiratory phase for each waveform; end-inspiratory transpulmonary pressure (Ptp-EI) and the end-expiratory transpulmonary pressure (Ptp-EE)—calculated as the difference between alveolar pressure and intrapleural pressure measured at the end of inspiration and end of expiration, respectively; end-inspiratory cardiac pressure (PEIC) and end-expiratory cardiac pressure (PEEC); peak inspiratory and expiratory pressure (PIP and PEP); FiO2.

For consistency, all measurements were performed by a single researcher trained in this technique. The primary endpoint of the bench study was to compare PEEP levels achieved with tHFTI versus nHFTI at identical flow rates. The main secondary endpoints were to evaluate the effects of nHFTI on Vt, FRC, Ptp, cardiac pressure, Peak airway pressure and FiO2.

Statistical analysis

Variables in this study were normally distributed and presented as mean ± SD. In the bench experiment, two-way analysis of variance (ANOVA) with repeated-measures was used to compare the parameters across different flow levels (10 to 80 L/min) and between tHFTI and nHFTI. Post hoc pairwise comparisons were performed using the Bonferroni correction. With nHFTI, PEEP and flow were fitted to the following power equation: PEEP = a×Flowb. The flow-PEEP curve was fitted using nonlinear regression, and the coefficient of determination (R2) was calculated. Correlation between PEEP and FRC was assessed using Spearman’s rank-order correlation; rs and the P value were reported. Statistical analyses were performed with SPSS 25.0 (IBM SPSS Statistics for Windows, Version 25.0. Armonk, NY, USA) and a two-tail P value < 0.05 was considered as statistically significant.

Results

Changes in the primary endpoint (PEEP, the positive end-expiratory pressure)

As the HFT flow rate increased, PEEP significantly increased with both tHFTI and nHFTI (P < 0.010). Compared with tHFTI, the nHFTI generated significantly higher PEEP at each flow rate (P < 0.010, Table 1, Fig. 3A).

Table 1 Primary and secondary endpoint parameters during high-flow tracheal oxygen in the bench experiment.

Parameter	HFTI	Flow rate (L/min)	
		10	20	30	40	50	60	70	80	P a	
PEEP
(cmH2O)	tHFTI	0.1 ± 0.0	0.3 ± 0.1	0.6 ± 0.1	1.0 ± 0.0	1.9 ± 0.1	3.1 ± 0.2	4.4 ± 0.1	5.7 ± 0.1	<0.001	
nHFTI	0.2 ± 0.1	0.9 ± 0.1	1.7 ± 0.1	3.4 ± 0.1	5.4 ± 0.1	7.3 ± 0.2	9.6 ± 0.2	12.7 ± 0.2	<0.001	
P *	0.004	<0.001	<0.001	<0.001	<0.001	<0.001	<0.001	<0.001		
FRC
(mL)	tHFTI	2,031.9 ± 1.3	2,039.4 ± 0.5	2,053.3 ± 1.1	2,069.2 ± 1.3	2,097.2 ± 2.3	2,145.6 ± 4.0	2,278.0 ± 2.5	2,465.0 ± 3.3	<0.001	
nHFTI	2,031.8 ± 0.7	2,054.3 ± 0.8	2,086.3 ± 1.1	2,155.0 ± 3.3	2,375.2 ± 2.3	2,610.0 ± 6.2	2,877.9 ± 4.0	3,282.8 ± 6.7	<0.001	
P *	0.759	<0.001	<0.001	<0.001	<0.001	<0.001	<0.001	<0.001		
Vt
(mL)	tHFTI	504.3 ± 8.2	502.3 ± 12.7	494.1 ± 2.3	489.9 ± 8.7	481.0 ± 11.7	475.9 ± 12.3	469.4 ± 12.8	459.8 ± 18.3	<0.001	
nHFTI	505.6 ± 5.4	496.2 ± 7.9	476.1 ± 7.8	454.2 ± 11.3	426.2 ± 11.3	403.1 ± 3.7	379.1 ± 6.1	343.7 ± 5.7	<0.001	
P *	0.679	0.211	<0.001	<0.001	<0.001	<0.001	<0.001	<0.001		
Ptp-EI
(cmH2O)	tHFTI	6.8 ± 0.2	6.9 ± 0.1	6.9 ± 0.1	7.0 ± 0.2	7.3 ± 0.2	7.6 ± 0.1	8.0 ± 0.0	8.6 ± 0.1	<0.001	
nHFTI	7.1 ± 0.2	7.4 ± 0.2	7.5 ± 0.1	7.9 ± 0.2	8.3 ± 0.1	8.9 ± 0.1	9.6 ± 0.2	11.4 ± 0.1	<0.001	
P *	<0.001	<0.001	<0.001	<0.001	<0.001	<0.001	<0.001	<0.001		
Ptp-EE
(cmH2O)	tHFTI	0.1 ± 0.1	0.3 ± 0.1	0.5 ± 0.1	1.1 ± 0.1	1.7 ± 0.1	2.9 ± 0.1	3.4 ± 0.1	4.0 ± 0.1	<0.001	
nHFTI	0.1 ± 0.1	0.6 ± 0.0	1.5 ± 0.1	2.9 ± 0.1	3.8 ± 0.1	4.6 ± 0.1	5.6 ± 0.1	6.7 ± 0.1	<0.001	
P *	0.500	<0.001	<0.001	<0.001	<0.001	<0.001	<0.001	<0.001		
PIP
(cmH2O)	tHFTI	−1.7 ± 0.1	−1.7 ± 0.2	−1.6 ± 0.2	−1.4 ± 0.1	−1.1 ± 0.2	−0.6 ± 0.1	0.6 ± 0.2	1.9 ± 0.1	<0.001	
nHFTI	−1.8 ± 0.2	−1.7 ± 0.1	−1.2 ± 0.3	−1.0 ± 0.2	0.7 ± 0.2	2.7 ± 0.2	4.6 ± 0.3	7.0 ± 0.2	<0.001	
P *	0.089	0.472	0.003	<0.001	<0.001	<0.001	<0.001	<0.001		
PEP
(cmH2O)	tHFTI	1.5 ± 0.1	1.9 ± 0.2	2.7 ± 0.1	3.3 ± 0.1	4.2 ± 0.1	5.5 ± 0.1	6.7 ± 0.3	8.4 ± 0.2	<0.001	
nHFTI	2.1 ± 0.2	3.2 ± 0.1	3.8 ± 0.4	5.4 ± 0.2	7.3 ± 0.4	9.0 ± 0.1	11.4 ± 0.2	14.9 ± 0.3	<0.001	
P *	<0.001	<0.001	<0.001	<0.001	<0.001	<0.001	<0.001	<0.001		
PEIC
(cmH2O)	tHFTI	−6.9 ± 0.1	−6.9 ± 0.1	−6.8 ± 0.1	−6.7 ± 0.1	−6.5 ± 0.1	−6.2 ± 0.1	−5.6 ± 0.1	−4.9 ± 0.1	<0.001	
nHFTI	−6.8 ± 0.1	−6.8 ± 0.1	−6.8 ± 0.1	−6.5 ± 0.1	−5.8 ± 0.2	−4.4 ± 0.1	−3.1 ± 0.1	−2.1 ± 0.1	<0.001	
P *	0.400	0.230	0.492	0.001	<0.001	<0.001	<0.001	<0.001		
PEEC
(cmH2O)	tHFTI	0.16 ± 0.01	0.20 ± 0.01	0.27 ± 0.01	0.37 ± 0.01	0.56 ± 0.02	1.21 ± 0.06	1.96 ± 0.08	2.79 ± 0.07	<0.001	
nHFTI	0.16 ± 0.01	0.28 ± 0.01	0.47 ± 0.03	1.17 ± 0.09	2.19 ± 0.11	3.35 ± 0.08	4.70 ± 0.11	6.64 ± 0.13	<0.001	
P *	0.310	<0.001	<0.001	<0.001	<0.001	<0.001	<0.001	<0.001		
FiO2
(%)	tHFTI	39.2 ± 0.7	43.6 ± 0.4	47.4 ± 0.5	48.7 ± 1.2	49.6 ± 1.1	50.4 ± 0.9	50.9 ± 1.1	51.1 ± 0.8	<0.001	
nHFTI	38.8 ± 0.7	43.6 ± 1.6	47.7 ± 0.8	49.0 ± 0.4	49.8 ± 0.4	50.6 ± 1.1	51.1 ± 0.2	51.2 ± 0.4	<0.001	
P *	0.132	0.939	0.274	0.478	0.489	0.532	0.561	0.741		
Notes.

FiO2, fraction of inspired oxygen; FRC, functional residual capacity; HFTI, high-flow tracheal interface; nHFTI, novel high-flow tracheal interface; PEEC, end-expiratory cardiac pressure; PEEP, positive end-expiratory pressure; PEIC, end-inspiratory cardiac pressure; PEP, peak expiratory pressure; PIP, peak inspiratory pressure; Ptp, transpulmonary pressure; Ptp-EE, end-expiratory transpulmonary pressure; Ptp-EI, end-inspiratory transpulmonary pressure; tHFTI, traditional high-flow tracheal interface; Vt, the tidal volume.

* Between tHFTI and nHFTI.

a Among flow rates.

Figure 3 Comparison of respiratory parameters between tHFTI and nHFTI across flow rates of 10–80 L/min.

(A) PEEP generated by tHFTI and nHFTI. nHFTI produces significantly higher PEEP than tHFTI at all tested flow rates. (B) FRC generated by tHFTI and nHFTI. nHFTI produces significantly higher FRC than tHFTI at all flow rates except 10 L/min. (C) Vt generated by tHFTI and nHFTI. Vt is lower with nHFTI than tHFTI at all flow rates except 10 and 20 L/min. (D) Ptp-EI generated by tHFTI and nHFTI. nHFTI produces significantly higher Ptp-EI than tHFTI at all tested flow rates. (E) Ptp-EE generated by tHFTI and nHFTI. nHFTI produces higher Ptp-EE than tHFTI at all flow rates except 10 L/min. (F) PIP generated by tHFTI and nHFTI. nHFTI produces significantly higher PIP than tHFTI at all flow rates except 10 and 20 L/min. (G) PEP generated by tHFTI and nHFTI. nHFTI produces higher PEP than tHFTI at all tested flow rates. (H) PEIC generated by tHFTI and nHFTI. nHFTI produces significantly higher PEIC than tHFTI at all flow rates except 10, 20, and 30 L/min. (I) PEEC generated by tHFTI and nHFTI. nHFTI produces higher PEEC than tHFTI at all flow rates except 10 L/min. (J) FiO2 generated by tHFTI and nHFTI. No significant difference in FiO2 was observed between the two interfaces. FiO2, fraction of inspired oxygen; FRC, functional residual capacity; HFTI, high-flow tracheal interface; nHFTI, novel high-flow tracheal interface; PEEC, end-expiratory cardiac pressure; PEEP, positive end-expiratory pressure; PEIC, end-inspiratory cardiac pressure; PEP, peak expiratory pressure; PIP, peak inspiratory pressure; Ptp, transpulmonary pressure; Ptp-EE, end-expiratory transpulmonary pressure; Ptp-EI, end-inspiratory transpulmonary pressure; tHFTI, traditional high-flow tracheal interface; Vt, tidal volume.

Changes in the secondary endpoint

Changes in functional residual capacity (FRC) and tidal volume (Vt):

With increasing HFT flow rate, both tHFTI and nHFTI led to a significant rise in FRC and a concurrent decline in Vt (P < 0.001). At all tested flow levels except 10 L/min, nHFTI produced higher FRC than tHFTI (P < 0.001), while Vt was lower with nHFTI except at 10 and 20 L/min (P < 0.001, Table 1, Figs. 3B and 3C). A strong and significant correlation was found between PEEP and FRC (rs = 1.0, P < 0.001).

Changes in end-inspiratory transpulmonary pressure (Ptp-EI) and end-expiratory transpulmonary pressure (Ptp-EE):

Ptp-EI and Ptp-EE increased significantly with flow for both tHFTI and nHFTI (P < 0.001). Compared with tHFTI, the nHFTI generated significantly higher Ptp-EI at all flow rates (P < 0.001) and higher Ptp-EE except at 10 L/min (P < 0.001, Table 1, Figs. 3D and 3E).

Changes in peak inspiratory pressure (PIP) and peak expiratory pressure (PEP):

A flow-dependent increase in PIP and PEP was observed with both tHFTI and nHFTI (P < 0.001). Compared with tHFTI, the nHFTI generated significantly higher PIP except at 10 and 20 L/min (P < 0.010) and higher PEP at all flow rates (P < 0.001, Table 1, Figs. 3F and 3G).

Changes in end-inspiratory cardiac pressure (PEIC) and end-expiratory cardiac pressure (PEEC):

PEIC and PEEC showed a significant increase with rising HFT flow rate, regardless of interface type (P < 0.001). The nHFTI produced higher PEIC than tHFTI except at 10, 20, and 30 L/min (P < 0.001) and higher PEEC except at 10 L/min (P < 0.001, Table 1, Figs. 3H and 3I).

Changes in fraction of inspired oxygen (FiO2):

FiO2 significantly increased with rising HFT flow rate (P < 0.001), with no significant difference between tHFTI and nHFTI (P > 0.05, Table 1, Fig. 3J).

Discussion

At the same flow rate, the novel high-flow tracheal interface (nHFTI) generated higher PEEP, peak inspiratory pressure (PIP), peak expiratory pressure (PEP), end-inspiratory transpulmonary pressure (Ptp-EI), end-expiratory transpulmonary pressure (Ptp-EE), end-inspiratory cardiac pressure (PEIC) and end-expiratory cardiac pressure (PEEC) compared to the traditional high-flow tracheal interface (tHFTI). Additionally, it also led to an increase in functional residual capacity (FRC). In contrast, fraction of inspired oxygen (FiO2) remained relatively stable with both interfaces, while tidal volume (Vt) was slightly lower with nHFTI.

PEEP is the positive pressure that remains in the airways at the end of the respiratory cycle (end of exhalation) and is greater than the atmospheric pressure in mechanically ventilated patients. The application of PEEP was first described in the 1930s (Poulton & Oxon, 1936; Barach, Martin & Eckman, 1938) and came into common use for treating acute respiratory distress syndrome (ARDS) in the 1960s (Ashbaugh et al., 1967).

Few studies have explored the airway pressures delivered by HFTO and how they vary with different gas flow rates. In a benchtop experiment, Thomas et al. (Thomas, Joshi & Cave, 2021) found that HFT can generate measurable and variable PEEP despite the open system design. At flow rates of 40, 50, and 60 L/min, PEEP levels were approximately 0.3, 0.5 and 0.9 cmH2O, respectively. In a single-center randomized crossover study, Natalini et al. (2019) compared standard oxygen therapy with HFTO in tracheostomized patients. Peak and mean expiratory pressures with HFTO at 50 L/min were 1.8 (1.4–2.2) cmH2O and 1.2 (1–1.5) cmH2O, respectively.

The study also compared HFTO and high-flow nasal cannula oxygen (HFNO) in a subgroup of five patients who underwent tracheostomy decannulation, and found that compared with HFNO, changes in expiratory pressure were significantly smaller with HFTO. At 50 L/min flow, the median peak expiratory pressure was 5.1 cmH2O with HFNO and 1.8 cmH2O with HFTO; the mean expiratory pressure was 3.9 cmH2O with HFNO and 1.5 cmH2O with HFTO. This within-patient comparison of HFTO and HFNO provided a unique opportunity to highlight the contribution of upper airway resistance to positive-pressure generation during HFNO.

We hypothesized that therapeutic PEEP could be generated by adding an exhalation valve to the exhalation port of the traditional interface to simulate nasal resistance during exhalation. In our benchtop study, the nHFTI was modified by incorporating an exhalation valve, thereby mimicking the nasal resistance during expiration and resulting in the generation of significant PEEP. In contrast to HFNO, which generate minimal intrathoracic PEEP (Rubin et al., 2014), the nHFTI is designed to intentionally modulate expiratory resistance to achieve therapeutic PEEP levels. At 80 L/min, maximum PEEP reached 12.7 ± 0.2 cmH2O, while at 50 L/min, PEEP was 5.4 ± 0.1 cmH2O, which was comparable to the maximum PEEP (5.7 ± 0.1 cmH2O) generated by tHFTI at 80 L/min.

PEEP generated by nHFTI in our bench study was approximately 3.4 cmH2O at a flow rate of 40 L/min, which was comparable to the pharyngeal pressure reported in adult patients receiving HFNO (Groves & Tobin, 2007; Parke, McGuinness & Eccleston, 2009; Corley et al., 2011; Parke, Eccleston & McGuinness, 2011). Previous HFNO studies showed that the airway pressure was determined by flow, with a linear (Groves & Tobin, 2007; Parke, Eccleston & McGuinness, 2011) or quadratic (Kumar, Spence & Tawhai, 2015; Luo et al., 2017) relationship between pressure and flow rate. In our bench model with nHFTI, PEEP and flow rate fit well with a power function: PEEP = 0.002 × Flow1.986 (R2 = 0.998, see Fig. 4), suggesting a much steeper increase in PEEP at higher flow rates.

Figure 4 Relationship between PEEP and flow rate during nHFTI.

PEEP increases FRC, thereby improving oxygenation through recruitment of collapsed alveoli and reduction of intrapulmonary shunt. In heart failure, PEEP may enhance hemodynamics by reducing venous return, decreasing left-ventricular afterload, and lowering transmural pressure. Excessive PEEP, however, can impair cardiac output and cause alveolar overdistension, potentially leading to volutrauma or barotrauma.

Our research confirmed that nHFTI produced higher PEEP than tHFTI. Additionally, the nHFTI slightly increased external cardiac pressure at both end-inspiration and end-expiration (PEIC and PEEC, particularly PEEC), primarily due to the higher effective PEEP. This addresses limitations of traditional interfaces for heart failure management. Based on our findings, the nHFTI can generate a PEEP of approximately 3–7 cmH2O at gas flow rates between 40 and 60 L/min. The increase in PEEP and external cardiac pressure observed with nHFTI was mild under benchtop conditions. These findings suggest that the device may produce hemodynamic effects that are less likely to impair cardiac output or induce hypotension, which are potential concerns with higher levels of PEEP in clinical settings.

Pulmonary complications such as alveolar collapse and atelectasis are common in critically ill patients and are associated with adverse outcomes, including prolonged mechanical ventilation, increased risk of ventilator-associated pneumonia, and extended intensive care unit and hospital lengths of stay. The main physiological consequence of alveolar collapse and atelectasis is decreased FRC. FRC is the volume of gas remaining in the lungs at the end of expiration during spontaneous breathing at atmospheric pressure (Leith & Brown, 1999). It represents the equilibrium point for the spring out forces of the chest wall and the collapsing tendency of the lung (Rahn et al., 1946) and is a major determinant of pulmonary oxygenation capacity. The term end-expiratory lung volume (EELV) is used to indicate FRC, if PEEP is applied.

A previous observational study indicated that HFNO increases EELV and alveolar ventilation compared with low-flow oxygen therapy (Corley et al., 2011). In this study, the nHFTI significantly increased FRC compared with tHFTI, and additionally, increased FRC was highly correlated (Spearman’s correlation rs = 1.0, P < 0.001) with increased PEEP.

We also observed that an increase in FRC was accompanied by a decrease in Vt, which can be attributed to increased expiratory resistance. While this study was conducted under simulated conditions, the observed increase in FRC suggests potential benefits in maintaining alveolar patency and improving ventilation-perfusion (V/Q) matching, which may support oxygenation in clinical settings. Although Vt decreased slightly—possibly affecting alveolar ventilation—the magnitude of this change was considerably smaller than the increase in FRC. For example, at a flow rate of 50 L/min, the nHFTI increased FRC by 278 mL compared to tHFTI, while decreasing Vt by only 54.8 mL on average. These results suggest that the new device may offer a favorable balance between lung volume recruitment and tidal volume preservation, which could support better oxygenation compared to the traditional device.

The difference between the alveolar pressure and the intrapleural pressure is known as transpulmonary pressure (Ptp). Ptp is the force that determines the expansion and contraction of the alveoli and is also key to maintaining alveolar patency at the end of expiration.

Our study demonstrated that increasing gas flow rates led to higher Ptp-EI and Ptp-EE, with a more pronounced increase observed with nHFTI compared to tHFTI. Elevated Ptp-EI reduces inspiratory work, while the concomitant rise in Ptp-EE increases expiratory work. These counteracting effects largely offset each other, resulting in only modest changes to the overall work of breathing.

In this study, Ptp-EI reached 11.4 ± 0.1 cmH2O at a flow rate of 80 L/min, a level unlikely to cause alveolar overdistension and subsequent alveolar injury. Maintaining Ptp-EE ≥0 cmH2O helps preserve alveolar patency at end-expiration. Clinically, Ptp-EE should be maintained at 0–5 cmH2O to prevent end-expiratory alveolar collapse while minimizing the risk of barotrauma and shear stress-related injury. In this model, Ptp-EE with nHFTI at flow rates of 40–60 L/min fell within this recommended range (0–5 cmH2O).

Increasing flow rate leads to a concomitant rise in peak airway pressure. Compared to tHFTI, the nHFTI increased both PIP and PEP. These elevated pressures, however, remains well below thresholds for pressure-induced lung injury. Notably, in ARDS management, it is the plateau pressure—rather than peak pressure—that is used to assess the risk of ventilator-induced lung injury, with current guidelines recommending maintenance below 30 cmH2O.

In this crossover study, we found that the nHFTI, compared with tHFTI, mitigates the negative swing in airway pressure during inspiration (allowing for a limitation of the negative swing in inspiratory airway pressure). Importantly, reducing excessive negative inspiratory pressure swings in airway (and pleural) pressure helps prevent negative pressure pulmonary edema, which can cause lung injury and impair oxygenation (Yoshida et al., 2017). Concurrently, the increase in expiratory pressure with nHFTI may contribute to higher EELV, reduced shunt fraction, improved lung mechanics, and better oxygenation (Corley et al., 2011; Parke, Bloch & McGuinness, 2015; Mauri et al., 2017a; Mauri et al., 2017b).

Theoretically, HFT delivers warmed and humidified gas with a relatively constant oxygen concentration. However, at low flow rates, significant room air entrainment during inspiration results in a delivered FiO2 substantially lower than the set value, compromising the accuracy of oxygen delivery and clinical monitoring. As flow increases, the delivered oxygen concentration rises due to reduced entrainment of room air. This study demonstrated that at flow rates of 40 L/min and above, the delivered FiO2 closely matched the preset concentration, with no statistically significant difference between the two.

In recent years, HFTO has been increasingly utilized as a supportive therapy in tracheostomized patients challenging to wean from invasive mechanical ventilation. Clinical studies on HFTO, however, remains limited to case reports (Mitaka et al., 2018; Ramachandran, Jha & Sircar, 2021; Vadi, Phadtare & Shetty, 2021) and studies investigating HFTO surrounding the process of decannulation (Hernández Martínez et al., 2020; Lionello et al., 2023).

Our findings contribute to the limited evidence on HFTO use in tracheostomized patients, a practice that appears to be increasing (Corley et al., 2017). Conventional HFTO delivery resembles HFNO administration with an open mouth, during which the extra expiratory resistance vanished, and the PEEP effect disappeared (Parke, McGuinness & Eccleston, 2009; Parke, Eccleston & McGuinness, 2011; Luo et al., 2017). In the bench study of nHFTI incorporating the exhalation valve, we observed a PEEP effect of 3.5–7.5 cmH2O at flow rates of 40–60 L/min, with a significant increase in FRC and a slight decrease in Vt. These findings highlight the efficacy of the new interface.

We must acknowledge several study limitations. As a benchtop investigation, we evaluated PEEP effects within a simulated respiratory system; consequently, the findings may not directly translate to clinical practice. A further limitation is the absence of gas humidification, which deviates from standard clinical care and may alter airway surface conditions—such as mucus viscosity and epithelial function—thereby reducing the physiological relevance of the results. Additionally, although the controlled environment of this model eliminates many clinical confounders and enables precise measurement of respiratory parameters, it inherently lacks patient-level variability. This means critical physiological responses—such as cardiovascular effects and safety risks related to auto-PEEP or dynamic hyperinflation—could not be assessed. Therefore, the speculative nature of clinical translation warrants caution in interpreting these results.

The study demonstrated that nHFTI increased PEEP while simultaneously reducing tidal volume, highlighting a potential trade-off between alveolar recruitment and ventilation efficiency. While this underscores the importance of selecting an appropriate inspiratory flow rate tailored to individual patient condition, such clinical decisions must ultimately be informed by future in vivo studies that rigorously evaluate both therapeutic efficacy and patient safety.

Conclusions

Compared to tHFTI, the nHFTI generated a modest level of PEEP, increased FRC and slightly reduced Vt at flow rates of 40–60 L/min. This physiological profile suggests potential for maintaining lung expansion and improving cardiac function in heart failure. Further in vivo studies are needed to evaluate the clinical impact of nHFTI-mediated PEEP augmentation and determine whether it can lead to improved clinical outcomes, particularly in patients requiring prolonged mechanical ventilation during the weaning process.

Supplemental Information

Supplemental Information 1 PEEP and other respiratory physiological indicators generated by the n HFTI at 10L/min

PEEP, positive end-expiratory pressure; nHFTI, new high-flow tracheal interface;

Supplemental Information 2 PEEP and other respiratory physiological indicators generated by the n HFTI at 30L/min

PEEP, positive end-expiratory pressure; nHFTI, new high-flow tracheal interface;

Supplemental Information 3 PEEP and other respiratory physiological indicators generated by the n HFTI at 70 L/min

PEEP, positive end-expiratory pressure; nHFTI, new high-flow tracheal interface;

Supplemental Information 4 PEEP and other respiratory physiological indicators generated by the tHFTI at 80 L/min

PEEP, positive end-expiratory pressure; tHFTI, traditional high-flow tracheal interface;

Supplemental Information 5 PEEP and other respiratory physiological indicators generated by the tHFTI at 20 L/min

PEEP, positive end-expiratory pressure; tHFTI, traditional high-flow tracheal interface;

Supplemental Information 6 PEEP and other respiratory physiological indicators generated by the n HFTI at 60L/min

PEEP, positive end-expiratory pressure; nHFTI, new high-flow tracheal interface;

Supplemental Information 7 PEEP and other respiratory physiological indicators generated by the tHFTI at 30 L/min

PEEP, positive end-expiratory pressure; tHFTI, traditional high-flow tracheal interface;

Supplemental Information 8 PEEP and other respiratory physiological indicators generated by the tHFTI at 10 L/min

PEEP, positive end-expiratory pressure; tHFTI, traditional high-flow tracheal interface;

Supplemental Information 9 PEEP and other respiratory physiological indicators generated by the tHFTI at 70 L/min

PEEP, positive end-expiratory pressure; tHFTI, traditional high-flow tracheal interface;

Supplemental Information 10 PEEP and other respiratory physiological indicators generated by the n HFTI at 40L/min

PEEP, positive end-expiratory pressure; nHFTI, new high-flow tracheal interface;

Supplemental Information 11 PEEP and other respiratory physiological indicators generated by the tHFTI at 50 L/min

PEEP, positive end-expiratory pressure; tHFTI, traditional high-flow tracheal interface;

Supplemental Information 12 PEEP and other respiratory physiological indicators generated by the tHFTI at 40 L/min

PEEP, positive end-expiratory pressure; tHFTI, traditional high-flow tracheal interface;

Supplemental Information 13 PEEP and other respiratory physiological indicators generated by the n HFTI at 20L/min

PEEP, positive end-expiratory pressure; nHFTI, new high-flow tracheal interface;

Supplemental Information 14 PEEP and other respiratory physiological indicators generated by the n HFTI at 80L/min

PEEP, positive end-expiratory pressure; nHFTI, new high-flow tracheal interface;

Supplemental Information 15 PEEP and other respiratory physiological indicators generated by the n HFTI at 50L/min

PEEP, positive end-expiratory pressure; nHFTI, new high-flow tracheal interface;

Supplemental Information 16 PEEP and other respiratory physiological indicators generated by the tHFTI at 60 L/min

PEEP, positive end-expiratory pressure; tHFTI, traditional high-flow tracheal interface;

Additional Information and Declarations

Competing Interests

Author Contributions

Patent Disclosures

Data Availability

Anna Hou, Song Mi and Liming Zhang have applied for a patent on the novel high-flow tracheal interface that has been already granted by the national intellectual property administration in China (ZL 2024 2 0280842.6). Apart from the above, the authors declare that they have no competing interests.

Anna Hou conceived and designed the experiments, performed the experiments, analyzed the data, prepared figures and/or tables, authored or reviewed drafts of the article, and approved the final draft.

Song Mi performed the experiments, prepared figures and/or tables, and approved the final draft.

Fengwei Jiao performed the experiments, prepared figures and/or tables, and approved the final draft.

Liming Zhang conceived and designed the experiments, authored or reviewed drafts of the article, and approved the final draft.

Zhaohui Tong conceived and designed the experiments, authored or reviewed drafts of the article, and approved the final draft.

The following patent dependencies were disclosed by the authors:

ZL 2024 2 0280842.6.

The following information was supplied regarding data availability:

The raw data are available in the Supplemental Files.

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
