# Peer review of "Design and simulation-based evaluation of a novel tracheostomy high-flow therapy device interface: a preclinical study"

_PeerJ, doi:10.7717/peerj.20445_

## Round 0.1 · original submission · Major Revisions

· Academic Editor

Major Revisions

**Language Note:** The review process has identified that the English language must be improved. PeerJ can provide language editing services - please contact us at [email protected] for pricing (be sure to provide your manuscript number and title). Alternatively, you should make your own arrangements to improve the language quality and provide details in your response letter. – PeerJ Staff

Reviewer 1 ·

Basic reporting

The authors present the results of a benchtop simulation experiment where PEEP was measured as a function of gas flow rate in simulated high flow tracheostomy using traditional and new tracheostomy interfaces.

The language needs only minor modifications.

The article is appropriately structured.

While there are many secondary endpoints, they are all reported clearly and appropriately.

Experimental design

This is original research within the scope of the journal.

The question that the authors asked was answered with the experimental setup and measures. This experiment is a new approach to asking a question which has been answered previously for traditional tracheostomy interfaces, and asks a new question regarding a new tracheostomy interface design.

Appropriate statistical tests appear to have been used.

I've tried a few times and don't understand how you delivered dry gas to the HFLS while excluding the humidifier. This aspect needs fixing to make it more understandable. A bigger figure with coloured arrows showing the flow of dry and humidified gas would be helpful.

Validity of the findings

(i) I don't believe the title matches the findings. You have designed a new HFTI and found that it behaved as you predicted in a simulator. This is not verification of a device designed for clinical use.

(ii) I am very concerned at the lack of a clear secondary release valve in your setup. PEEP measured at the highest flows at > 12 cm is high. On reading your paper I am not confident that the release valve in the new HFTI does not have the potential to block with catastrophic consequences. As a clinician this is the first aspect that comes to mind after reading your paper. Fixing this with a clear description of how potential for blockage to the expiratory port has been minimised and release valve setups in any HFTI circuit would be very important for any future revisions.

(iii) The results are scientifically valid and well stated

(iv) In general there is a need to tone down statements on the clinical applicability of your device. PEEP has uses. Your device increased PEEP in a simulated model. You need in vivo and safety data before venturing into comments on clinical applicability.

Additional comments

Minor comments:

Line 62, - "could generate a modest level of PEEP", adding "a".
The sentence from line 83-87 could be deleted without losing any value in the manuscript.
The sentence in line 96 may be true, but you have no evidence for this. It would be true if it read "scarcity of the literature may lead to hesitation...",

Line 141- lack of humidification of gas delivered to the HFLS represents a difference from the clinical situation and thus a limitation of the experiment.

Line 203 - Fix "During with..."

Line 245 - You can delete this sentence without losing any meaning from the manuscript. It is in my view overreach to comment on the clinical applicability of your device at this stage.

Lines 293-306 could be collapsed into some very short statements about PEEP, such as it increases FRC and thus can improve oxygen exchange and is of use in managing cardiac failure due to effects on venous return, afterload and FRC. This is all accepted and known by anyone with an interest in the application of PEEP.

Point (iv) from section 3 is particularly pertinent to the last sentence of the conclusions.

·

Basic reporting

The manuscript is well structured and logically organized. The introduction clearly outlines the rationale for the study, citing relevant literature and identifying a clinical gap: the inability of traditional high-flow tracheostomy oxygen (tHFTO) to generate meaningful airway pressure.
Figures and tables are informative and clear. The results are presented with sufficient statistical detail. However, the manuscript would benefit from professional editing for medical English. While the authors do an admirable job communicating technical concepts in a second language—no small feat—the readability and precision of the writing would be improved by revision. The grammar, sentence structure, and phrasing occasionally hinder clarity and may limit accessibility for an international audience.
Examples of sentences that would benefit from more precise medical English editing include:
• Starting each paragraph of the results with “The results showed that…”. Consider starting the sentence with the word that follows “that” in each of those paragraphs.
• “…positive patient-centered outcomes” can be changed to “salutary outcomes” or “improved outcomes”
• No need to repeat the patent number in line 122 since it has already been mentioned in line 102.
I believe the manuscript adheres to PeerJ structural guidelines and provides appropriate referencing throughout. A declaration of competing interests by the authors is included, as the authors clearly mention their intellectual property rights to the nHFTO device.

Experimental design

This is an original benchtop investigation aimed at evaluating a novel high-flow tracheostomy interface (nHFTI) designed to augment airway pressure by increasing expiratory resistance, compared to the traditional interface (tHFTI). The experimental design is sound, employing a randomized crossover setup using a high-fidelity lung simulator to assess multiple flow-dependent respiratory parameters across both interfaces.
The rationale is clearly stated, and the study design is appropriate for an in vitro bench model. Methods are described in sufficient detail to allow replication. The authors have carefully documented device specifications, interface construction, flow rates, and measured variables, including PEEP, FRC, Ptp, and cardiac pressures.

Validity of the findings

The findings are internally consistent and statistically robust, as one would have expected from a rigorously conducted bench study. The novel interface appears to generate significantly higher PEEP and other beneficial respiratory parameters compared to the traditional interface, especially at flow rates ≥40 L/min.
That said, the authors should temper their clinical enthusiasm. This remains an early preclinical benchtop study, and although it offers compelling physiologic insights, translation to human subjects remains completely speculative at this point. The authors acknowledge this limitation, but it deserves greater emphasis, particularly regarding patient safety and hemodynamic effects.
One issue must be addressed is the fact that the authors spent a considerable amount of time introducing and discussing high flow nasal oxygen (HFNO). The similarities between HFNO and HFTO end in high flow; they are vastly different modalities with different purposes. HFNO works by significantly decreasing work of breathing by providing conditioned (heated and humidified gas) at high flow and high speed across the nasopharyngeal resistors while also washing off carbon dioxide from the anatomical dead space. Positive pressure generation during HFNO is negligible and clinically insignificant. The studies cited by the authors to imply that HFNO generates significant PEEP (references 9 to 12) demonstrated elevation of higher expiratory pressure in the pharynx, not in the intrathoracic cavity (pleural space). HFNO is predicated on an “open system”, meaning that the cannula should occupy less than 50%V of the cross sectional area of the nostrils to allow sufficient gas to leak around it. Again, the purpose of HFNO is not to pressurize the intrathoracic compartment (generate PEEP) but to decrease work of breathing through bypassing the nasopharyngeal resistor and washing off carbon dioxide from the anatomical dead space. The authors should clarify that HFNO primarily increases pharyngeal pressure and reduces work of breathing, not necessarily by generating meaningful intrathoracic PEEP. I recommend referencing Rubin et al. (Pediatr Crit Care Med. 2014;15(1):1–6), who demonstrated reduced effort of breathing but minimal actual PEEP (<1 cm H₂O) in pediatric subjects using HFNC. Overstating HFNO’s PEEP effect undermines the novelty of the new interface.
In fact, the authors would be best served by removing the references to HFNO altogether from the manuscript and focus on what is really important here, the effect of their novel device on HFTO. This can be easily done by deleting the entire first paragraph of the introduction (lines 73 to 80), and lines 258 to 261 and 297 to 298 in the discussion.
The paragraph from line 303 to 306 should also be deleted as it is a gross oversimplification of cardiopulmonary interactions.

Additional comments

Congratulations on an important and well-executed study. The device concept is novel, and the bench data are compelling.

If afforded by the editor the opportunity to revise your manuscript, consider doing so with assistance from a native English speaker or professional medical editor to improve clarity and technical language.

The claim that HFNO generates PEEP should be reframed. You may consider clarifying that most HFNO studies report pharyngeal pressure changes rather than clinically significant airway PEEP. Referencing Rubin et al. (2014) may be helpful. Or better yet, delete all mention of HFNO and focus on what this study really is: a good bench study of HFTO.

While your data demonstrate a important physiologic advantage of the nHFTI, please ensure that all clinical implications are appropriately qualified as hypothetical pending in vivo validation.

---

## Round 0.2 · Minor Revisions

· Academic Editor

Minor Revisions

Reviewer 1 ·

Basic reporting

I had no substantive issues with this aspect of the paper at first revision, and I find it improved with this revision.

Experimental design

The issue I raised re: gas flow has been rectified.

Validity of the findings

Thank you for the alteration of the title, which now reflects the nature and findings of your experiment.

I appreciate your noting of the safety issue with respect to the potential blockage of the expiratory port. While I appreciate the calibre of the nHFTI expiratory port, the valve described in lines 531/2 remains of concern for me in this aspect. I commend you for considering a redundant high-pressure blow-off feature in the future - it would, in my view, be most welcome to any potential market.

Thank you for revising the paper generally to better reflect the preclinical nature of your findings.

Additional comments

Thank you for the minor revisions undertaken.

The parentheses in the second-to-last paragraph of the discussion, starting line 2500, disrupt the flow and might be deleted.

I think this paper is pretty close to publishable - the discussion is still a little wordy, but I appreciate the author's point that the readership may need some more context around the physiologic effects of PEEP.

·

Basic reporting

This is a vastly improved version of the manuscript. The authors adequately addressed all my concerns and suggestions.

Experimental design

-

Validity of the findings

-

---

## Round 0.3 · accepted · Accept

· Academic Editor

Accept

Thank you for revising your manuscript to address the concerns of the reviewers, both of whom are satisfied with the revisions you have made in response to their comments. The manuscript is now ready for acceptance.

Reviewer 1 ·

Basic reporting

As previous, no issues

Experimental design

As above

Validity of the findings

As above

Additional comments

I would like to thank the authors for their respectful and thoughtful responses to all issues raised. I wish them all the best with further development of their technology.